# Detecting Subtle Cognitive Impairment in Patients with Parkinson’s Disease and Normal Cognition: A Novel Cognitive Control Challenge Task (C3T)

**DOI:** 10.3390/brainsci13060961

**Published:** 2023-06-16

**Authors:** Karmen Resnik Robida, Vida Ana Politakis, Aleš Oblak, Anka Slana Ozimič, Helena Burger, Zvezdan Pirtošek, Jurij Bon

**Affiliations:** 1University Rehabilitation Institute, University of Ljubljana, SI-1000 Ljubljana, Slovenia; vida.politakis@ir-rs.si (V.A.P.); helena.burger@ir-rs.si (H.B.); 2University Psychiatric Clinic Ljubljana, SI-1260 Ljubljana, Slovenia; ales.oblak@psih-klinika.si; 3Department of Psychology, Faculty of Arts, University of Ljubljana, SI-1000 Ljubljana, Slovenia; anka.slanaozimic@ff.uni-lj.si; 4Department of Physical Medicine and Rehabilitation, Faculty of Medicine, University of Ljubljana, SI-1000 Ljubljana, Slovenia; 5Department of Neurology, University Medical Centre Ljubljana, SI-1000 Ljubljana, Slovenia; 6MRD Center for Active and Healthy Ageing, University of Ljubljana, SI-1000 Ljubljana, Slovenia; 7Department of Psychiatry, Faculty of Medicine, University of Ljubljana, SI-1000 Ljubljana, Slovenia

**Keywords:** stable cognitive control, flexible cognitive control, cognitive control challenge task, Parkinson’s disease, task switching

## Abstract

Patients with Parkinson’s disease (PD) often show early deficits in cognitive control, with primary difficulties in flexibility and relatively intact stable representations. The aim of our study was to assess executive function using an ecologically valid approach that combines measures of stability and flexibility. Fourteen patients without cognitive deficits and sixteen comparable control subjects completed a standardized neuropsychological test battery and a newly developed cognitive control challenge task (C3T). We found that the accuracy of C3T performance decreased with age in healthy participants and remained impaired in PD patients regardless of age. In addition, PD patients showed significantly lower overall performance for cognitive control tasks than healthy controls, even when they scored in the normal range on standardized neuropsychological tests. PD Patients responded significantly faster than healthy control subjects regarding flexible cognitive control tasks due to their impulsivity. Correlations showed that the C3T task targets multiple cognitive systems, including working memory, inhibition, and task switching, providing a reliable measure of complex cognitive control. C3T could be a valuable tool for characterizing cognitive deficits associated with PD and appears to be a more sensitive measure than standardized neuropsychological tests. A different assessment approach could potentially detect early signs of the disease and identify opportunities for early intervention with neuroprotective therapies.

## 1. Introduction

Recent cohort studies have provided valuable insight into the substantial impact of cognitive impairment on the development of Parkinson’s disease (PD) [1]. While motor symptoms are well known, it is increasingly recognized that non-motor symptoms such as sleep disturbances, depression, anxiety, olfactory impairment, and mild cognitive deficits often occur in the prodromal stage of PD [2]. Research has even shown a correlation between non-motor symptoms, with the severity of olfactory impairment increasing with the depth of executive dysfunction [3]. These findings underscore the importance of considering non-motor symptoms, particularly executive functions, as important predictors of disease progression [2]. Subtle behavioral changes may be observed years before clinical diagnosis, and the early-onset cognitive pattern differs from the cognitive pattern observed in advanced disease with PD dementia [4,5,6]. It is characterized by the mild impairment of executive abilities, which also distinguishes it from other neurodegenerative diseases such as Lewy body dementia (LBD), in which cognitive impairments, particularly of attention and executive functions, are more pronounced than in PD [7]. Similarly, fronto-temporal dementia (FTD), which is also characterized by early changes in behavior, language, and executive function impairments, appears to differ from PD, in which executive function impairments are typically milder and more related to frontal-subcortical circuit dysfunction [8], and it also differs from Alzheimer’s disease (AD), which primarily affects memory early in the disease process [9,10,11].

These findings highlight the potential benefits of the early detection and assessment of specific cognitive deficits in PD and offer valuable diagnostic and therapeutic advantages. However, further research is needed to fully realize these benefits, with a particular focus on developing more sophisticated and comprehensive assessments of executive function [1].

Assessing early executive dysfunction in PD is challenging because it is a multifaceted phenomenon. Mirabella [12] and Hampshire & Sharp [13] jointly emphasize the complex nature of executive functions and their underlying neural networks. Mirabella’s study explores the basis of inhibitory control and highlights the distributed nature of executive functions by emphasizing the interplay between the prefrontal cortex, basal ganglia, and other brain regions. Building on this, Hampshire and Sharp emphasize that executive functions involve a dynamic network of brain regions beyond the frontal lobes. They emphasize the importance of considering the interaction and functional connectivity between different brain regions for a comprehensive understanding of executive functions. According to Diamond [14,15] and Miyake et al. [16], there are three key executive functions: First, inhibitory control refers to the ability to suppress automatic or prepotent responses and to inhibit irrelevant information or impulses. This includes resisting distractions, suppressing impulsive behaviors, and focusing on relevant stimuli or tasks. Second, working memory involves the temporary storage and manipulation of information in the mind. It is responsible for storing and processing information relevant to ongoing tasks and enables individuals to mentally process and update information in real time. Working memory capacity is important for tasks that require the storage of multiple pieces of information, such as following instructions, solving problems, and making decisions. Third, cognitive flexibility refers to the ability to adapt to changing demands or situations and to switch between different cognitive strategies, mental sets, or perspectives. This includes the ability to shift attention, adjust behavior, and switch between different rules or strategies, as needed. The other, more complex executive functions such as attention, self-control, and volition derive from those [17,18]. 

Executive function encompasses a broader range of cognitive processes involved in the higher-level control and regulation of behavior. Cognitive control is a specific aspect of executive function that refers to the processes involved in regulating and controlling cognitive operations. Cognitive control refers to the set of cognitive processes and mechanisms involved in regulating and directing our thoughts, actions, and behavior in order to achieve specific goals [14,15,16,19]. According to the dual mechanisms of the cognitive control account [20], goal attainment can be supported by proactive or reactive cognitive control processes. Proactive cognitive control refers to the establishment and maintenance of goal representations in advance of relevant target stimuli based on an external or internal cue. In contrast, reactive cognitive control involves the activation or retrieval of goal representations at the time they are needed. With regard to the stability of task goals, their achievement can be supported by cognitive control mechanisms that enable their stable maintenance or flexible change. When task goals remain constant over time, stable cognitive control is required to establish a set of cognitive processes and relevant information necessary for the efficient completion of an ongoing task while protecting them from interference by irrelevant stimuli and events [21]. Conversely, when current goals change dynamically due to environmental demands, flexible cognitive control becomes necessary. It enables individuals to switch between different task sets and mental operations, select and integrate relevant information in novel ways, and prioritize relevant information while adapting to changing environmental demands [17,19,22,23,24,25].

While, at first glance, proactive control may be primarily associated with the stable mode of operation and reactive with flexible change, a stable environment may actually promote reactive cognitive control, because in a stable environment, the task set is easier to recall and reactivate at the time of execution, and there is less potential interference. Conversely, a changing environment requires active reconfiguration, involving either the reactivation of previously used task sets or the establishment of new task sets and the inhibition of previously relevant task sets. The efficient reconfiguration and prevention of interference from previously active task sets may therefore involve proactive cognitive control to a greater extent. In summary, reactive/proactive cognitive control can be applied regardless of whether ongoing task goals remain constant over time or change from one event to another.

The stable representation of rule sets and stimuli must be maintained over time, typically measured by working memory tasks or distractibility judgments, and these tasks have been shown to involve the tonic dopaminergic stimulation of D1 receptors in PFC [26]. Second, successful executive performance requires the flexibility to direct attention to novel stimuli and to apply different rules based on new information. This is usually measured by set-switching tasks and is associated with the phasic dopaminergic stimulation of striatal D2 receptors [27].

However, the relationship between dopamine deficiency and cognitive control in PD is complex and not fully understood [28]. The effects of dopamine deficit in the dorsal striatum on deficits in set-shifting are task-dependent and not consistently correlated [28]. Deficits in cognitive control in PD include performance impairments for tasks involving the suppression of irrelevant information or the updating of working memory and deficits in switching between tasks [6,29,30,31,32,33,34]. Deficits in task switching are characterized by difficulties in flexibility or an increased tendency to perseverate [35,36] and could be due to different cognitive processes of stability and flexibility that are differentially affected by the dopaminergic state [33,37,38,39,40,41]. Moreover, one of the most striking cognitive impairments in PD is impaired inhibitory control [42,43]. Inhibition has been identified as a sensitive measure of disease progression, with different patterns observed at different stages of the disease [43,44]. Specifically, in the earliest stage of PD (Hoehn and Yahr (HY) stage 1), reactive inhibition is impaired, whereas in HY 2, both reactive and proactive inhibition deficits are observed [43,44]. These deficits in inhibitory control worsen as the disease progresses beyond HY 2.5. Executive functions are elaborated by large networks that dynamically adapt to the contextual factors in which a person is embedded. This networked view underscores the need to study executive functions beyond localized brain regions and encourages the exploration of network-level dynamics to capture the complexity of multilayered cognitive processes [12,13]. Finding metrics sensitive enough to distinguish abnormal cognitive patterns is critical for detecting early signs of disease progression and potential opportunities for early intervention with neuroprotective therapies. As Burgess and Stuss [45] pointed out, “the absence of a deficit on executive tests does not necessarily mean that there is no problem” (p. 764).

Behavioral assessments primarily rely on a component-based approach to assess cognitive control, which may limit the applicability of the measures used in practice [11,46]. In the past, it has been shown that behavioral tasks may be too simplistic [47]. The limitations of most neuropsychological tests are that they usually assess only the function of a single executive function [48]. Therefore, the proposed shift in the general assessment of executive functions from a brain region approach to a brain system approach could contribute to a better understanding of executive functions [45]. Tasks that are more ecologically valid and can assess cognitive control in ambiguous and complex situations with unclear task rules have been developed in recent years with these methodological issues in mind [25,49,50] and have been shown to effectively measure a wide range of cognitive abilities.

To our knowledge, this type of multimodal approach has not yet been performed in patients with PD. Thus, in this study, we use a novel task that provides a more holistic measure of executive functioning. 

We aim to investigate the differences between stable and flexible cognitive control in PD patients without cognitive impairment. The Cognitive Control Challenge Task (C3T) has been previously validated in a large sample of healthy individuals across the lifespan and provides information on the development of cognitive control. It appears that stable and flexible cognitive control are two distinct cognitive processes with different developmental trajectories [25]. The C3T structure allows for the measurement of the preparation time, reaction time, and performance accuracy for task sets divided into two modes: the stable task mode and flexible task mode. In the stable task mode, the same rules are used continuously, which allows for observing the time required to create a new task set and update the set on subsequent trials. In the flexible task mode, the rules change from trial to trial, which provides insight into the time required to switch between previously learned complex task sets. The separation and fixed order of task modes allow for the observation of different types of training. Improvements in the stable task mode provide information about task set acquisition, optimization, and execution progress, while the flexible task mode reveals improvements in switching between task sets.

The aim of this study is to investigate the ability of the C3T to detect subtle cognitive changes that are not detectable by standardized neuropsychological tests, to distinguish between stable and flexible task modes, and to assess whether patients with PD show deficits in the flexible task mode compared to healthy controls. The hypothesis is that patients with PD will show increased errors in the flexible task mode, whereas they will perform similarly to healthy controls in the stable task mode.

## 2. Materials and Methods

### 2.1. Participants

A total of 17 participants with PD (5 women) and 16 healthy controls (6 women) were recruited for the study, and they were matched in terms of gender, age, and education level. All participants were native Slovene speakers. The inclusion criteria were based on a detailed neuropsychological assessment (see Section 2.3), and only participants who were within the normative sample on cognitive assessments (z-score > −1 SD) were included in the subsequent analysis (N = 14). PD patients diagnosed with mild cognitive impairment (MCI) according to the Diagnostic Criteria for Mild Cognitive Impairment in Parkinson’s Disease of the Movement Disorder Society Task Force (N = 1; [51]) and PD patients who had moderate or severe impairment (z score > −2.00) in one or more cognitive domains were excluded from the study (N = 2). Other exclusion criteria included medical conditions that might impair cognitive performance, an inability to perform the C3T experimental paradigm after the training session, and physical, visual, or auditory impairments that would prevent participation in the tasks. Finally, a total of 14 patients and 16 control subjects were included in the analysis of behavioral data. The patients were recruited at a tertiary care University Rehabilitation Institute during their first hospitalization, with a mean duration of illness of 5.4 years ± 2.8 years. Idiopathic PD was diagnosed before admission to the hospital by a senior neurologist, who confirmed the clinical diagnosis in all our patients and ruled out other possible differential diagnoses. We did not include patients with other types of parkinsonism, and none of our subjects reported subjective cognitive complaints. In addition, our exclusion criteria were such that all participants with other suspected neurodegenerative diseases were excluded. All patients were taking medication to treat PD and were receiving stable treatment with a dopaminergic agonist or levodopa. The clinical data for each PD patient and the group mean are shown in Table 1. None of the patients were taking antipsychotics or sedatives. We studied patients in their best “on-phase state” at the same time of day for each part of the study.

The study protocol was approved by the National Medical Ethics Committee of the Republic of Slovenia (protocol code 92/08/17, date of approval 20 February 2018). All subjects gave written informed consent according to the Declaration of Helsinki.

### 2.2. Study Design

Each participant completed a series of standardized neuropsychological tests in one or two sessions after signing an informed consent form. The initial cognitive tests were performed in the clinical setting (clinical psychologist’s office), whereas the C3T task was performed in the Movement Disorders Laboratory of the Department of Neurology. 

### 2.3. Cognitive Assessment

The participants were assessed by a clinical psychologist using a comprehensive neuropsychological test battery that focused primarily on cognitive domains related to cognitive control and fluid intelligence. First, the mental processing speed was examined using a publicly available version of the Trail making test (TM A and B condition; [52]) with an additional sensorimotor control condition (TM C). Next, executive functions were assessed with the standardized Tower of London Test (TOL; [53]) and the Stroop Color-Word Test (SCWT; [54]). Verbal, visual, and working memory were assessed with a verbal learning test (VLT), the Rey–Osterrieth Complex Figure Test (RCFT; [55]), a working memory test that included a forward and backward digit span, alphabetic letter span, and even and odd digit span (WM), and an automated computerized version of the visual and spatial span based on the original test by Unsworth et al. [56]. Verbal fluency was assessed using a verbal fluency test with lexical, semantic, and category switching tasks. Finally, verbal (TVS; [57]) and nonverbal intellectual abilities (Raven’s Standard Progressive Matrices (SPM) test; [58]) were assessed. We used a validated Slovenian version of the language-dependent cognitive tests.

The tests were always administered in the same order but did not have to be completed in the same session if the participant felt tired. If the tests were divided into two sessions, they were administered at the same time on the following day. 

### 2.4. Cognitive Control Challenge Task

The C3T [25] is a novel computerized testing procedure that allows for a more accurate examination of cognitive control. In the validation study, the C3T assessment was shown to correlate with standardized cognitive tests and to measure three key executive functions: inhibition (of non-target stimuli), working memory (in stable task mode), and flexibility (in flexible task mode). More specifically, C3T measures preparation time, reaction time, and response accuracy for task sets divided into stable and flexible modes. In the stable task mode, where consistent rules are applied, it allows for the study of the time required to construct and refresh task sets in subsequent trials. In the flexible task mode, the rule changes from trial to trial, providing valuable insight into the timing of switching between previously learned complex task sets. This clear separation and fixed order of task modes allow for the observation and analysis of different training modes. Progress in the stable task mode reflects the acquisition, optimization, and execution of task sets, while the flexible task mode reveals improvements in the ability to switch between task sets. The test itself is lengthy, taking about an hour to an hour and a half (depending on the participant’s reaction time, since it is self-paced). Once they begin the computer-based assessment, participants are continuously exposed to a two-part task within one trial. In the first part, the general rule they must follow is presented. The rule appears on the screen (Figure 1A). The first line describes where they have to focus their attention (e.g., search for a living organism). Then, they must decode the rule about the target stimuli (e.g., search for the larger living organism). The third line tells them which button to press to give their answer (e.g., if the larger living organism is presented on the right, press the right button; if it is presented on the left, press the left button). After 4 s, the instructions disappear. The screen goes blank for 6 s, and participants must decode the rule and retain the instructions in their working memory. This period is called the maintenance phase. In the second part, four stimuli are presented simultaneously on the screen and in the headphones. Each of the stimuli has a different modality, so suppressing irrelevant stimuli and focusing attention on the relevant stimuli are critical to responding successfully to a task. Figure 1B shows an example of such a stimulus presentation. In the left earpiece, a participant (he) hears a word spoken by a female voice called “house”. At the same time, in the right ear, he hears the barking of a dog. The participant must concentrate exclusively on a living organism, i.e., the barking, and suppress the unnecessary information (house). At the same time, two visual stimuli appear on the right and left sides of the computer screen, with the fixation point in the center. The stimuli are, again, two modalities (a written word—a bee—and a picture of an object—a ladder). Again, based on the previously given instructions, he must direct his attention to a living organism, i.e., a bee. Now, he must remember the rule that the target is a living organism that is larger, i.e., if he compares a bee and a dog, the correct answer is a dog. Since the barking was heard on the right side (headphones), he will press the right button to answer a task correctly. 

The rules presented are either consistent over a block of trials (stable task mode) or change randomly from trial to trial (flexible task mode). In each mode, there are 12 blocks of 16 trials, resulting in 192 trials in the stable mode and 192 trials in the flexible mode. As shown in Figure 1C, the stable mode consists of blocks of 16 trials with the same rule (e.g., green represents a rule: alive, bigger, left... right), and the same rule is presented 16 times (however, the stimuli presented are always different, so the responses are unique in each trial). The structure of the flexible mode contains constantly changing rules in each trial, so the participant has to decode a new rule after each response. Counterbalancing was not applied because participants had to respond to the stable task mode first, which allowed them to understand and learn the principles of the task. In the second part of the test, the rules changed at irregular intervals, and participants had to respond flexibly to the changing rules.

The temporal structure consisted of a 2 s baseline, a 4 s presentation of the rule (the preparation phase with the establishment of the mental set), a 6 s maintenance phase, and a self-paced response time. Improvement over trials in the stable mode provides information on task acquisition, optimization, and execution, while the flexible mode provides specific information on task switching improvement (see Figure 1C for details).

### 2.5. Statistical Analyses

The independent-samples *t*-test or Mann–Whitney test was used to ensure that patients and control subjects were indeed matched for age, educational level, and performance on standardized neuropsychological tests. False discovery rate (FDR) correction for multiple comparisons was used (*p*-value-adjusted), as implemented in the R function p.adjust.

Pearson correlations were calculated to examine the correlations of the C3T measures with the results of the standardized neuropsychological tests. To account for multiple comparisons, the *p*-values within each sample were adjusted and reported using Benjamini and Yekutieli’s [59] FDR correction.

For the statistical analysis of the behavioral data (i.e., the performance measurements on the C3T), a custom script was written using the R statistical programming language [60] and RStudio [61]. The mixed ANOVA was run for the between-group (patients vs. controls) and within-subjects (flexible vs. stable) design. Two separate analyses assessed the performance accuracy (PA) and reaction times (RT). Both performance measures were obtained by averaging across all trials of each participant. We first examined whether the data met the parametric assumptions of mixed ANOVA. We used the identify_outliers function from the R package rstatix to automatically detect outliers. No outliers were detected for RT. Three outliers were detected for PA. Although they were not flagged as extreme outliers, we removed them from the analysis. We tested the normal distribution of the data using the Shapiro–Wilk test and visual inspection of the QQ plot. Levene’s test was used to test the homogeneity of variance, and Box’s M test was used to test the homogeneity of covariance. Finally, the Mauchly test was used to check for sphericity. Parametric assumptions were met for both PA and RT. We used ANOVA to test for differences between groups and within subjects. When a significant interaction was detected, we performed post hoc tests. Bonferroni correction for multiple comparisons was applied to each test. We examined the effect of the group at each time point and the effect of the condition at each level of the grouping variable using one-way ANOVA. We conducted pairwise comparisons between grouping levels using a pairwise t-test. Finally, we used a pairwise t-test for pairwise comparisons between time points at each level of the grouping variable. If no significant interaction was found, we performed multiple paired t-tests for the condition variable, ignoring the group. We then conducted multiple paired t-tests for the grouping variable, ignoring the condition. Linear regression modeling was performed to clarify an emerging research question.

## 3. Results

We studied the patient group with a mean age of 67.79 years (SD = 8.23) and an educational level of 13.77 years (SD = 3.38) and the control group with a mean age of 68.94 years (SD = 7.63) and an educational level of 13.69 years (SD = 3.38). We used an independent-samples t-test to determine that there were no differences between the two groups. Participants were the same in terms of age (t(28) = 0.398, *p* = 0.694) and years of education completed (t(28) = −0.333, *p* = 0.742). 

### 3.1. Standardized Neuropsychological Assessment Performance

To test the differences in performance regarding various cognitive tests between the patient and control groups, the independent-samples *t*-test or the Mann–Whitney test was used. The results show that the performance for all tests except the verbal learning test (VLD) is comparable between groups. The results of the neuropsychological test battery are summarized in Table 2. 

### 3.2. Comparison of Performance on Standard Cognitive Tests and the C3T Task

We examined the relationship between measures of the C3T task and other cognitive control tests. Specifically, we examined correlations between performance accuracy (PA) and reaction times (RT) in the stable and flexible task modes with patient and control scores on tests of working memory span (WM), trail making (TM), long-term memory (VLT, RCFT), Tower of London (TOL), Stroop task (incongruent score), verbal fluency (VF), and intellectual ability (SPM, TVS).

Our analysis of the results, shown in Table 3, revealed that there were no correlations between the standardized neuropsychological tests and C3T task performance in the patient group. In the group of healthy participants, we found significant correlations between the C3T measures and the measures of the cognitive performance tests. Speed of mental processing (TM) was negatively correlated with performance accuracy measures in both the stable and flexible conditions, whereas performance on the Stroop task was positively correlated with performance accuracy measures in the incongruent condition. Working memory (WM) and verbal memory (VLT) scores were positively associated with lower error rates in the stable condition and shorter reaction times. In addition, we found a significant correlation between verbal memory and intellectual abilities (SPM, TVS) and C3T performance accuracy in the stable task mode.

Finally, we found that C3T performance measures were correlated with measures of verbal and nonverbal intellectual ability. Specifically, healthy participants with higher intellectual abilities were able to solve the task more accurately (Table 3).

### 3.3. C3T Performance Accuracy 

First, we focused on the distribution of the performance accuracy (PA) as a function of age. In healthy individuals, a slow decline in PA is observed, whereas in PD patients, the performance accuracy is less affected by age (Figure 2). 

Multiple linear regression was used to test whether group and age had a significant effect on PA. Two separate models were used, one for PA in the stable condition and another for PA in the flexible condition. The overall regression for the stable condition was statistically significant (R2 = 0.097, F(2, 23) = 4.901, *p* = 0.017). The group was found to significantly predict the value of PA (β = 0.487, *p* = 0.010), while age had no significant effect on PA (−0.233, *p* = 0.195). The overall regression for the flexible condition was statistically significant (R2 = 0.034, F(2, 23) = 6.977, *p* = 0.004). It was found that the group significantly predicted PA (β = 0.484, *p* = 0.007), as did age (β = −0.364, *p* = 0.037).

Second, we focused on examining the distribution of accuracy for both groups to test how successful participants were in solving the task. The scatter plot of the mean accuracy per participant across all trials in the C3T task shows that both groups solved the task better than chance in both conditions (red dotted line; Figure 3A). In the stable condition, the two groups are approximately equally successful at solving according to the density distribution, but in the flexible condition, the control participants are more successful than the patients. Further analysis confirmed significant differences in solving accuracy.

For PA, the mixed ANOVA showed no significant effect of the interaction between the group and condition (F (1, 25) = 1.829, *p* = 0.188). However, we observed a significant main effect of the group (F (1, 25) = 10.309, *p* = 0.004) and condition (F (1, 25) = 7.827, *p* = 0.010). The pairwise comparison between flexible and stable conditions was significant (*p* = 0.009). The pairwise comparison between the control and patient groups was also significant (*p* < 0.001). The difference in PA is summarized in Figure 3B.

### 3.4. C3T Response Times

The time it took participants to solve the task was relatively stable across trials, suggesting that a single exposure to the rule was sufficient to learn the task principles. On subsequent trials, participants stably maintained the coded cognitive configuration, as evidenced by their consistently short preparation times (Figure 4A). The analysis also revealed significant differences in solving time between the patients and controls.

For RT, the mixed ANOVA showed a significant effect of interaction between the group and condition (F (1, 28) = 4.210, *p* = 0.050). Considering the Bonferroni-adjusted *p*-value, the simple main effect of the group was significant in the flexible condition (*p* = 0.032) but not in the stable condition (*p* = 0.36). Pairwise comparisons show that the mean RT was significantly different between the patients and control subjects in the flexible condition (*p* = 0.0163) but not in the stable condition (*p* = 0.18). There was a statistically significant effect of the condition on the mean of RT in the control group (*p* = 0.008) but not in the patient group (*p* = 1.0). In pairwise comparisons with paired t-tests, the mean of RT was statistically significantly different between flexible and stable conditions in the controls (*p* = 0.004) but not in the patients (*p* = 0.816). The difference in RT is summarized in Figure 4B.

## 4. Discussion

The aim of the present study was to use the C3T [25] as a measure of complex cognitive control in individuals with PD. One of the main advantages of the C3T task is its ability to detect subtle cognitive changes that are not detectable with standard neuropsychological tests. Cognitive control is impaired in people with PD [31]. We lack data on how quickly the changes can be detected, but we know from other studies that standard assessments of cognitive ability may fail to detect symptoms that can be subjectively reported and clinically observed [47]. It has been shown that executive deficits in PD are often a precursor to motor symptoms and are distinct from the more profound cognitive impairments that occur in PD dementia [4,5,6]. It also appears that cognitive control problems are not a consequence of PD deterioration, as we see in PD-related dementia, but an initial symptom of mild cognitive impairment (MCI) [9,10,11]. This suggests that the detection and assessment of cognitive control deficits at an early stage could have significant diagnostic and therapeutic benefits for PD. We have shown that the C3T can also discriminate between stable and flexible task modes, which, to our knowledge, has not been studied simultaneously in PD patients.

### 4.1. Performance Accuracy Declines with Age for Healthy Participants but Remains Impaired Regardless of Age in PD Patients

We observed a slow decrease in C3T task performance accuracy in healthy subjects, which is consistent with the previous results of the C3T task validation study [25]. On the other hand, we observed a consistent impairment of PA, regardless of age, in PD patients. Aging may lead to a decline in several cognitive abilities, including processing speed and short-term memory [62], as well as problem-solving skills and flexibility, such as the ability to change problem-solving strategies and switch between tasks [63,64]. In addition, other cognitive functions such as attention, spatial perception, and executive control may also be negatively affected by aging [65,66,67]. While cognitive decline progresses with age in healthy individuals, cognitive functions remain permanently impaired in PD patients but do not necessarily worsen with age but rather with disease stage [68]. One possible explanation for the lack of age-related cognitive decline in PD patients is that the disease exerts a “ceiling effect” on cognitive performance, i.e., cognitive functions are at a lower level compared with healthy individuals from the onset of the disease and do not decline further with age [68]. This phenomenon has been attributed to the selective vulnerability of specific neuronal networks in PD, which may lead to impairments in cognitive areas not normally affected by aging [69]. Another possible explanation for the lack of age-related cognitive decline in PD patients is related to the use of medications to treat the motor symptoms of the disease. Research suggests that dopaminergic medications may have a positive effect on cognitive function in PD patients, at least in the short term [4]. Dopamine replacement therapy has been associated with improvements in working memory, attention, and executive functions [70]. The negative effects of dopaminergic therapy on specific cognitive functions in PD patients have also been observed. They are probably related to the well-known inverted U-shape of the therapeutic window of dopaminergic therapy, i.e., they can be explained by the “overdose dopamine hypothesis” [28]. According to this hypothesis, equal doses of dopaminergic drugs improve motor and cognitive functions relying on depleted dorsolateral striatal circuitry but overdose the relatively unaffected circuitry of the ventral striatum and associated prefrontal areas, impairing cognitive functions that depend on these brain regions. In addition, research has shown that PD patients may exhibit compensatory changes in neuronal activity to maintain cognitive performance despite neurodegeneration. For example, studies have found increased activity in frontal cortical regions in PD patients during tasks requiring cognitive control, suggesting that these individuals may recruit additional neural resources to support cognitive function [34,35].

### 4.2. Overall Performance on the Cognitive Control Task Is Significantly Lower in Patients with PD Compared with That in Healthy Controls, Even When They Are within the Normal Range on Standardized Neuropsychological Assessments

There was no statistically significant interaction effect on PA between patients and controls and between flexible and stable conditions. However, we demonstrated that there was a statistically significant difference in PA both between healthy controls and patients and between flexible and stable conditions. In the patient group, we did not find a significantly lower performance in the flexible condition, as expected, but rather an overall poorer performance on the task for the PD patients. Healthy control subjects solved the task more accurately than the patients. In our study, we included only patients who fell within the normative range of cognitive performance (Table 1). We excluded patients who met the diagnostic criteria for mild cognitive impairment in Parkinson’s disease according to the Movement Disorder Society Task Force [51] and patients with moderate or severe cognitive impairment (z-score > −2.00). Our study demonstrated the potential diagnostic value of the C3T task, as it has a higher sensitivity in detecting subtle cognitive changes compared with standardized neuropsychological batteries. These results show that a multimodal assessment approach has higher ecological validity and can detect subtle changes in cognitive control in ambiguous and complex situations with unclear task rules [25,45,49,50] when initial comprehensive neuropsychological assessments targeting single frontal lobe functions show normative results.

Moreover, the PA was higher in healthy control subjects in the flexible condition than in the stable condition. The results suggest that the C3T task may be too easy for the healthy control subjects. The continuous improvement of PA in the flexible condition probably reflects better rule recognition and the effect of exercise on performance efficiency. The task should be designed to mix trials with repetitive and alternating rules rather than completely separating stable and flexible conditions to eliminate the practice effect.

### 4.3. Patients with PD Respond Significantly Faster Than Healthy Controls during the Flexible Cognitive Control Task

In our study, we observed significant differences in reaction time (RT) between healthy participants and patients with PD in the flexible condition. Healthy participants exhibited significantly longer RTs, even though they were already familiar with the task rules from the stable condition. These results suggest the presence of switching costs in healthy control subjects, which has also been observed in other traditional switching tasks [71,72]. Switching costs are the additional time and effort required to switch from one task to another or to shift attention between different aspects of a task. These costs are observed in many cognitive tasks that require participants to switch between multiple rules, goals, or task sets [73]. The presence of switching costs can be observed in different types of switching tasks, such as those that involve task rules or stimuli and those that require the use of different response modalities [73,74].

Switching costs do not occur in PD patients in the same way as in healthy individuals, which is due to changes in their cognitive control processes. PD is associated with a decrease in the function of the basal ganglia, which play a crucial role in cognitive control processes, including task switching [4]. Specifically, PD patients show lower basal ganglia activation during task switching compared to healthy controls [34]. Moreover, PD patients may have difficulty switching between different cognitive tasks due to the impaired fronto-striatal system involved in cognitive flexibility and executive control [75,76]. Consequently, the lack of typical switching costs in PD patients is most likely a result of difficulties in cognitive flexibility.

Patients with PD may not show prolonged reaction times and respond impulsively to the presentation of stimuli, which increases the likelihood of errors. According to Hlavatá et al. [47], individuals with PD-related impulse disorder (PD ICD) tend to prioritize avoiding harm and minimizing effort in mentally demanding situations. Advanced PD patients have impaired proactive control (e.g., [43]). They find it difficult to maintain long-term goals and overcome obstacles, and they may abandon their goals when faced with difficulties.

### 4.4. The C3T Task Engages Multiple Cognitive Systems in Healthy Subjects, Including Working Memory, Inhibition, and Task-Switching, and Provides a Valid Measure of Complex Cognitive Control

In this study, we examined the relationship between C3T task measures and other neuropsychological tests and provided further evidence for the utility of the C3T task in assessing cognitive control skills. We found that working memory plays a role in the accuracy of cognitive control. Moreover, as expected, performance on the Trail Making task (TM), which measures mental speed, was associated with more accurate C3T performance in both stable and flexible task modes. In addition, higher mental flexibility and the ability to switch between modalities in the Stroop task (incongruent word-color condition) were highly associated with performance accuracy in both task modes. We also observed that C3T performance measures were correlated with verbal and nonverbal intellectual abilities. In particular, participants with higher intellectual abilities were able to solve the C3T task more accurately, especially in the stable task mode, further supporting the task’s ability to measure one’s capacity to learn or understand concepts and information.

Overall, our results suggest that healthy individuals with a higher mental speed, working memory capacity, verbal memory, and mental flexibility show better performance on the C3T task. These results provide further evidence for the construct validity of the C3T task as a measure of cognitive control abilities.

It is noteworthy that the significant correlations were limited to the sample of healthy individuals. This could be due to the greater variability and associated deterioration in cognitive performance in PD patients compared with healthy individuals. Other authors have also demonstrated greater variability in performance for attention and working memory tasks in PD patients compared to healthy controls [77]. In addition, studies have shown that greater intraindividual variability in neuropsychological performance is associated with a higher risk of cognitive decline in people with PD [78]. The greater variability in performance could potentially explain why there are no correlations in a small sample of patients. A small sample size and greater variability in the data could reduce the power to detect statistically significant correlations, and further studies should therefore examine larger samples.

### 4.5. Limitations

A potential limitation of this study is the relatively small sample size, which may limit the generalizability of our results. Although statistical analyses revealed significant differences between the PD and control groups, it is possible that a larger sample size would have yielded more robust results. In addition, the small sample size may have limited our ability to control for potential confounding variables. Further studies with larger sample sizes and more diverse populations are needed to confirm and extend our findings.

## 5. Conclusions

Because the C3T task provides a more comprehensive assessment of task switching and cognitive control processes, it can be a valuable tool for characterizing cognitive deficits associated with PD. The task allows for a more accurate assessment of the ability to adapt and integrate individual cognitive control processes when confronted with challenging tasks in real-world scenarios and is a more sensitive measure for discriminating between the cognitive performance of PD patients and healthy adults compared to standardized neuropsychological tests.

## Figures and Tables

**Figure 1 brainsci-13-00961-f001:**
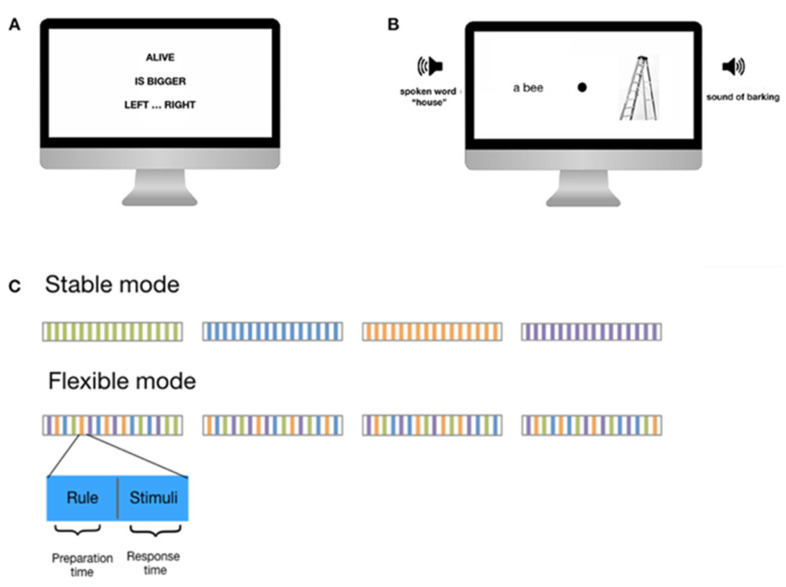
(**A**) An illustration of a single C3T trial is presented. First, a rule consisting of three elements is shown to the participant. (**B**) Then, a combination of auditory and visual stimuli is presented, to which the participant must respond by pressing the left or right keyboard button. (**C**) The stable mode structure consists of blocks of 16 repetitions of the same rule (green lines), and in the next block, a new rule applies (orange, etc.). The flexible mode structure consists of 16 randomly changing rules within a block. (Adapted with permission from [25]).

**Figure 2 brainsci-13-00961-f002:**
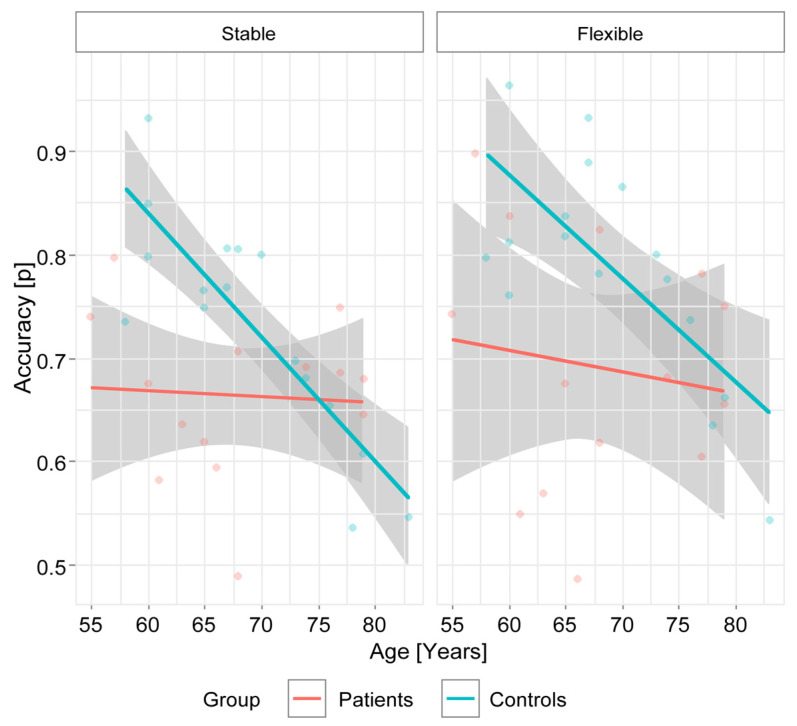
The performance accuracy according to age for patients and controls.

**Figure 3 brainsci-13-00961-f003:**
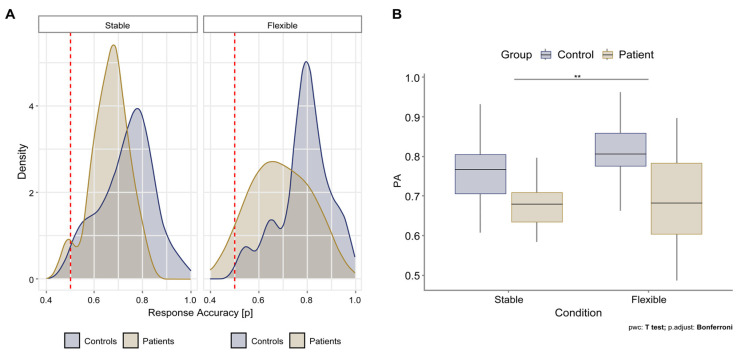
(**A**) Accuracy of C3T performance (PA) in both groups under stable and flexible conditions; (**B**) Difference in PA between stable and flexible conditions for patient and control groups. Asterisks denote significance levels of pairwise post hoc comparisons.

**Figure 4 brainsci-13-00961-f004:**
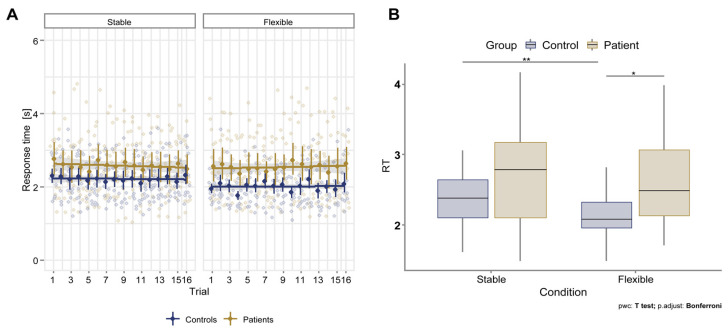
(**A**) Response time (RT) across trials in patients and healthy controls under stable and flexible conditions; (**B**) Difference in RT between stable and flexible conditions for patient and control groups. Asterisks denote significance levels of pairwise post hoc comparisons.

**Table 1 brainsci-13-00961-t001:** Clinical information of PD participants. Legend: PD = Idiopathic Parkinson’s disease; UPDRS-III = Unified PD Rating Scale part III; HY = Hoehn and Yahr stage; LEDD = levodopa equivalent daily dose; DA = dopamine agonist daily dose; SBP (mmHG) = systolic blood pressure; DBP (mmHG) = diastolic blood pressure. n/a = not available; SD = standard deviation.

Clinical Data	PD01	PD02	PD03	PD04	PD05	PD06	PD07	PD08	PD09	PD10	PD11	PD12	PD13	PD14	Mean	SD
Sex	f	m	m	m	m	m	m	f	m	f	m	m	f	f		
Age	65	61	55	57	60	79	77	79	74	68	77	63	68	66	67.8	7.9
PD duration (years)	2	6	2	2	1	6	7	5	10	8	4	10	3	6	5.4	2.8
UPDRS-III	16	24	40	33	26	28	28	30	42	30	49	31	43	49	33.5	9.4
HY	1	2	2.5	2	2	2.5	2.5	2	2.5	2.5	2.5	2.5	2.5	2.5	2.3	0.4
LEDD	400	300	1000	/	/	1200	300	300	200	400	250	900	200	800	520.8	337.6
DA (mg)	/	/	/	16	6	/	/	/	/	/	/	/	/	/	11.0	5.0
SBP (mmHG)	144	163	144	144	130	114	130	180	158	145	155	155	138	127	144.8	16.3
DBP (mmHg)	81	86	86	78	78	70	78	82	85	80	75	62	78	73	78.0	6.3
HR (bpm)	90	63	96	68	72	77	72	68	70	73	72	70	69	76	74.0	8.5

**Table 2 brainsci-13-00961-t002:** Participant properties on the cognitive assessment battery.

Cognitive Assessment	Patient Group	Control Group	*p*-Score	*p*-Score (Adjusted)
TMa	35.2 (12.8)	36.7 (16.3)	0.788	0.841
TMb	90.7 (34.5)	82.9 (41.6)	0.549	0.732
TMc	21.7 (8.6)	20.2 (8.9)	0.712	0.814
WM	14.1 (2.7)	14.9 (2.3)	0.393	0.629
WMss	2.8 (1.2)	3.2 (0.9)	0.173	0.624
WMvs	5.5 (1.6)	5.8 (1.2)	0.635	0.782
VLTld	7.8 (2.3)	10.5 (3.9)	0.046, *	0.624
RCFTld	58.5 (15.0)	62.1 (9.5)	0.466	0.678
TOL	109.4 (18.4)	103.3 (14.0)	0.351	0.624
TOLt	97.5 (11.9)	102.0 (9.7)	0.303	0.624
StrpInc	23.5 (5.9)	23.5 (7.0)	1.000	1.000
VFlex	25.2 (10.1)	31.5 (8.2)	0.086	0.624
Vfsem	33.7 (6.4)	37.9 (6.8)	0.117	0.624
VFsw	12.2 (2.7)	13.2 (2.8)	0.347	0.624
SPM	3.8 (1.8)	4.7 (2.0)	0.251	0.624
TVS	105.5 (13.9)	110.9 (12.8)	0.306	0.624

Legend: TMa = time to complete version A of the TMT task, TMb = time to complete version B of the TMT task, TMc = time to complete version C of the TMT task, WM = average number of correctly recalled numbers in sequence, reverse order, and alphabetical order, WMss = Working Memory Spatial Span, WMvs = Working Memory Visual Span, VLTld = number of recalled words in the Word Learning Test in the long-term recall condition, RCFTld = score on the Rey–Osterrieth Complex Figure Test long-term recall trial, TOL = accuracy on the Tower of London task, TOLt = time to complete the Tower of London task, StrpInc = STROOP task, incongruent condition, VFlex = average number of words in the lexical condition for the letters S, I, and T, VFsem = average number of words in the semantic condition for the categories animals and male names, VFsw = number of words in the switching condition between the categories fruits and furniture, SPM = accuracy on Raven’s matrices, TVS = verbal intellectual ability index. Asterisk denotes statistically significant difference. FDR correction for multiple comparisons, as implemented in the R function p. adjust, was used. * The results show that the performance for all tests except the verbal learning test (VLD) is comparable between groups.

**Table 3 brainsci-13-00961-t003:** Map of correlations. A map with values showing correlations between standardized cognitive tests and C3T measures (for (A) patients and (B) controls). Red colors indicate positive correlations, while blue colors indicate negative correlations. Stronger colors represent stronger correlations.

A/Patients					B/Controls			
	Stable		Flexible		Stable		Flexible	
	RA	RT	RA	RT	RA	RT	RA	RT
TMa	0.31	0.34	−0.06	0.26	−0.59	0.19	−0.52	−0.06
TMb	0.03	0.31	−0.30	0.19	−0.84	0.38	−0.77	0.08
TMc	−0.09	0.40	−0.35	0.28	−0.65	0.08	−0.57	0.01
WM	0.18	−0.03	0.29	−0.10	0.56	−0.56	0.35	−0.32
WMss	0.S0	−0.23	0.53	−0.30	0.33	0.11	0.27	−0.09
WMvs	0.27	−0.34	0.45	−0.32	−0.10	0.35	−0.22	−0.12
VLTId	−0.39	0.04	−0.12	0.13	0.63	−0.30	0.50	−0.13
RCFTId	0.24	−0.29	0.42	−0.34	0.29	−0.05	0.06	−0.33
TOL	0.31	0.21	0.07	0.21	0.35	−0.17	0.3	−0.50
TOLt	0.22	−0.05	0.29	0.05	0.00	0.01	−0.04	0.16
Strplnc	−0.25	0.09	−0.24	0.24	0.86	−0.49	0.76	−0.16
VFlex	0.04	−0.16	0.25	−0.08	0.49	−0.13	0.33	0.29
VFsem	−0.33	−0.33	−0.05	−0.09	0.31	−0.49	0.26	0.06
VFsw	0.12	−0.40	0.25	−0.26	0.23	0.09	0.22	0.43
SPM	0.06	0.18	0.00	0.08	0.69	0.06	0.63	0.20
TVS	0.42	0.02	0.19	0.01	0.68	−0.53	0.47	−0.38

Legend	*p* < 0.001	*p* < 0.01	*p* < 0.05	*p* < 0.05	*p* < 0.01	*p* < 0.001		

Legend: PA = accuracy on the C3T task; RT = reaction time on the C3T task; TMa = time to complete version A of the TMT task; TMb = time to complete version B of the TMT task; TMc = time to complete version C of the TMT task; WMss = Working Memory Spatial Span; WMvs = Working Memory Visual Span; WM = average number of correctly recalled numbers in sequence, reverse order, and alphabetical order; TOL = accuracy on the Tower of London task; TOLt = time to complete the Tower of London task; StrpInc = STROOP task, incongruent condition; VLTld = number of recalled words in the Word Learning Test in the long-term recall condition; RCFTld = score on the Rey–Osterrieth Complex Figure Test long-term recall trial; VFlex = average number of words in the lexical condition for the letters S, I, and T; VFsem = average number of words in the semantic condition for the categories animals and male names; VFsw = number of words in the switching condition between the categories fruits and furniture; SPM = accuracy on Raven’s matrices; TVS = verbal intellectual ability index.

## Data Availability

The data presented in this study are available on reasonable request from the corresponding author. The data are not publicly available due to privacy and ethical restrictions.

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
