# Peer review of "Detecting Subtle Cognitive Impairment in Patients with Parkinson’s Disease and Normal Cognition: A Novel Cognitive Control Challenge Task (C3T)"

_brainsci, 2023, doi:10.3390/brainsci13060961_

Round 1

Reviewer 1 Report

The file for review lacks a list of citations, so some lines of the manuscript (Introduction and Discussion) are difficult to fully analyze. At the same time, I would like to draw the attention of the authors to a number of points that can increase the value of the work.

1. The main problem in the diagnosis of PD is the differential diagnosis with a number of neurodegenerative diseases. For example, "Lewy bodies, Parkinson's disease, cerebrovascular disease, fronto-temporal dementia, Creutzfeldt-Jakob disease, HIV-related dementia, neurosyphilis, normal pressure hydrocephalus, and dementia due to exposure to current - septic substances (heavy metals, alcohol, other drugs), metabolic disorders or mental disorders". As well as Alzheimer's and multiple sclerosis in the early stages. Moreover, neuropsychological testing for initial and latent cognitive impairments will be similar.

2. In this regard, I recommend that the authors in the Introduction justify the choice of PD taking into account other similar diseases.

3. In the research was examined 17 patients with a short history of about 5 years. The main question is whether there was a PD? Therefore, in the Methods chapter, it is necessary to expand the information on the clinical criteria by which the diagnosis of PD was made. The only accurate method is the PET-scan, but due to the high cost, it is not often done. However, if there were patients with such a study, this can be confirmed.

4. Fig. 1. - technical note. The main text split fig. and signature?

5. The quality of all figures is not high. But fig. 5 need to improve the resolution!

6. In the Discussion, taking into account the lack of a list of references. Recently, there are works with factor characteristics of cognitive functions in PD and with PET-scan. It is advisable to take this into account in the new version of the manuscript.

7. List of references!

Reviewer 2 Report

The paper by Robida et al. aims to demonstrate the validity of a tool, the Cognitive Control Challenge Task (C3T), to measure executive functions in Parkinson’s disease (PD) patients.

It is difficult to fully understand what the C3T assesses and interpret the results. In addition, several crucial data need to be included, and there are considerable gaps in the Introduction and Discussion. Many concepts need to be properly described or supported by the literature. Finally, English could be of better quality. As it is, the paper is not publishable. In the following, I will provide a few comments.

MAJOR POINTS

1.     ENTIRE PAPER. As it is written, executive functions seem to be implemented solely by the frontal (Line 35)/prefrontal (Line 78) lobes. By far, this is wrong. Such complex and multifaceted functions are elaborated by large networks which change according to the context in which a person is embedded. See Mirabella (2014) for an example of the underpinnings of inhibitory control (see also Hampshire & Sharp, 2015).

2.     ENTIRE PAPER. There is an ill-definition of executive functions and cognitive control. According to Diamond (2013, 2020) and Miyake et al. (2000), there are three core executive (or cognitive) functions: inhibitory control, working memory, and set-shifting/cognitive flexibility. The other more elaborate executive functions derive from these. Such distinction is very useful for avoiding confusion. Self-control, volition, attention (Line 35), and planning (Line 156) are different and more complex constructs with respect to core executive functions. Global processing speed is not at all an executive function (Line 156).

3.     ENTIRE PAPER. Strictly related to the above point is the fact that is absolutely unclear what the C3T assess. The task is introduced on line 88 without any explanation (please delete lines 86-88, as they are meaningless and misleading). What distinctive features enable the authors to state that this is the ‘first task to assess both stable and flexible cognitive control’? What is the definition of stable versus flexible cognitive control? Do the authors believe that the C3T measure set-shifting/cognitive flexibility? If this is so, they must clearly state this. What about the other core executive functions (inhibition and working memory)? In the Methods (line 166-on), the C3T task is explained, but the description and the related figure (1) are difficult to interpret. How are the rules shown to the participants? Surely not, as shown in figure 1A. How many questions at a time a participant has to respond to? The legend of figure 1 is even more confusing. As this task is supposed to be the core of the paper, it must be described in detail, and the authors should describe what they would like to measure by using it.

  1. INTRODUCTION. Lines 55-67. Here there are several theoretical errors. First, the relationship between dopamine deficit and cognitive control is largely unclear (Cools, 2019). None was able to find clear-cut relationships between deficit in set-shifting and dopaminergic deficit in the dorsal striatum because these deficits are highly dependent on the task at play (Cools, 2019). Second, one of the hallmarks of PD is impaired inhibitory control (Gauggel et al., 2004; Mirabella et al., 2017). Relevantly for the paper’s topic, inhibition has been taken as sensitive outcome measure for disease progression, as it has been shown that in the earliest stage (H&Y1), reactive but not proactive inhibition is impaired (Di Caprio et al., 2020), while in H&Y2 PD patients the impairment is impact also proactive inhibition (Mirabella, 2017). Both inhibitory control types further deteriorate in more advanced stages (>H&Y 2.5). This paragraph must be completely re-written

5.     METHODS. Paragraph 2.1. Here some crucial data are missing. Clinical data (e.g., H&Y stage, onset of the disease, LEDD, MDS-UPDRS-III) must be reported for each PD participant, and the average data should be provided in a Table. Were patients under stable treatment with L-dopa, dopamine agonists, or L-Dopa and dopamine agonists? What is the procedure leading to scoring a participant ‘within the normative sample on the cognitive assessments’? It is unclear. In addition, 14 patients are a small sample.

6.     METHODS Line 162-165. If counterbalancing was not applied, and patients were tested first in the ‘stable task mode’ and then in the ‘flexibly task mode’, a learning effect is likely to occur, weakening the results’ value. Why was this procedure adopted?

7.     RESULTS. Lines 227-230. Provide the results of statistical tests about the age and years of education of PD patients versus healthy controls. By looking at figure 3, it seems that controls were older than patients. This would be a fatal error.

8.     RESULTS. Lines 249 After correction for multiple comparisons, the VLD test was not significant. Please correct and delete the column ‘p-score’ of Table 1. By the way, change ‘p-score’ to ‘p-value’.

9.     RESULTS. Figure 4B and 5B. Delete the statistical results and report the effect sizes in the main text.

10.  RESULTS. Figure 5A does not make any sense, and it is also unreadable.

I will stop here, but I have several criticisms also about the Discussion.

MINOR POINTS

1.     The acronyms are very badly defined. Many of them are repeatedly defined several times in the paper, e.g., C3T (Line 9,88,151,152…by the way, the meaning of this acronym is unclear). Performance accuracy sometimes is defined with PA (Line 206) some other times with RA (line 290 and Figure 4). Parkinson’s Disease is labeled as PD on line 96 and again on line 356, but not on lines 104 and 111. There are many more mistakes.

2.     Given the results provided in Table 1, figure 2 is useless. Please remove it.

3.     Line 335 what is the ‘solving time’?

4.     Line 346 It should be figure 5, not 6

References

11.            Cools, R. (2019). Chemistry of the Adaptive Mind: Lessons from Dopamine. Neuron, 104(1), 113-131. https://doi.org/10.1016/j.neuron.2019.09.035

12.            Di Caprio, V., Modugno, N., Mancini, C., Olivola, E., & Mirabella, G. (2020). Early-Stage Parkinson's Patients Show Selective Impairment in Reactive But Not Proactive Inhibition. 35(3), 409-418. https://doi.org/10.1002/mds.27920

13.            Diamond, A. (2013). Executive functions. Annual Review of Psychology, 64, 135-168. https://doi.org/10.1146/annurev-psych-113011-143750

14.            Diamond, A. (2020). Executive functions. Handbook of Clinical Neurology, 173, 225-240. https://doi.org/10.1016/b978-0-444-64150-2.00020-4

15.            Gauggel, S., Rieger, M., & Feghoff, T. A. (2004). Inhibition of ongoing responses in patients with Parkinson's disease. J Neurol Neurosurg Psychiatry, 75(4), 539-544. https://doi.org/10.1136/jnnp.2003.016469

16.            Hampshire, A., & Sharp, D. J. (2015). Contrasting network and modular perspectives on inhibitory control. Trends in Cognitive Sciences, 19(8), 445-452. https://doi.org/10.1016/j.tics.2015.06.006

17.            Mirabella, G. (2014). Should I stay or should I go? Conceptual underpinnings of goal-directed actions. Frontiers in Systems Neuroscience, 8, 206. https://doi.org/10.3389/fnsys.2014.00206

18.            Mirabella, G., Fragola, M., Giannini, G., Modugno, N., & Lakens, D. (2017). Inhibitory control is not lateralized in Parkinson's patients. Neuropsychologia, 102, 177-189. https://doi.org/10.1016/j.neuropsychologia.2017.06.025

19.            Miyake, A., Friedman, N. P., Emerson, M. J., Witzki, A. H., Howerter, A., & Wager, T. D. (2000). The unity and diversity of executive functions and their contributions to complex "Frontal Lobe" tasks: a latent variable analysis. Cognitive Psychology, 41(1), 49-100. https://doi.org/10.1006/cogp.1999.0734

As I have already stated, English must be improved.

Reviewer 3 Report

The manuscript (Manuscript ID: brainsci-2380845) entitled “Detecting Subtle Cognitive Impairment in Patients with Parkinson's Disease and Normal Cognition: A Novel Cognitive Control Challenge Task (C3T)”, aims to assess executive functions using cognitive control task (C3T) in patients with Parkinson's disease (PD) compared to healthy controls. The Manuscript is well written, newsworthy and clear to understand, but requires major revisions.

Main weaknesses of this Manuscript are the low number of PD patients and controls enrolled and the absence of important disease information such as UPDRS, disease duration, Hoehn and Yahr, and the levodopa equivalent daily dose.

Specific comments:

As indicated above the number of PD patients and controls for this study is very low and Authors should increase the number of subjects enrolled in their study.

Line 31 Authors should indicate the following reference as regards executive dysfunction in the PD prodromal stage also in the Introduction and Discussion: Solla et al. Biology 2023, 12,112. https://doi.org/10.3390/biology12010112

Authors should indicate a single section for the statistical analyses.

Line 194 Please indicate false discovery rate for the acronym of FDR at the first use.

Line 195 Authors should delete the double bracket.

Line 231 Probably, it should be more appropriate "to test the differences".

Lines 316 and 334 Authors should delete the double brackets.

Lines 361-362 Authors should consider that a previous study indicated a significant association between olfactory dysfunction and deficit in executive functions. In addition, the cognitive deficits may be related to more severe stages of the disease.

Minor editing of English language is required

Round 2

Reviewer 1 Report

The manuscript in this form can be recommended for publication.

Author Response

Thank you for ypur support and approval of the manuscript.

Reviewer 2 Report

After the first round of review, the paper has been improved, and, relevantly, now the task is well described. However, there are still several issues that must be fixed before I will endorse the paper.

  1. The most important one is the terminology that the authors applied. Throughout the manuscript, they keep referring to stable and flexible modes. However, this differs from the way, or the correct terminology, used to describe the operating modes of cognitive functions. Braver (2012) stated that cognitive control functions either in reactive or proactive modes (see also,  Mirabella, 2023). Reactive control reflects a mechanism exploited when needed, i.e., when an abrupt change of behavior is required, as in the case of what the authors define as ‘flexible mode’ (i.e., from time to time, participants are instructed to behave in a certain way). By contrast, the proactive mode involves a set of processes that allow people to maintain goals in a sustained manner despite distractions and interferences. I strongly suggest referring to this model. By the way, lines 112-116 seem to describe exactly reactive and proactive modes. In fact, the authors seem to assess the person’s ability to operate global shifts of executive functions, passing from a proactive (stable) to a reactive mode (flexibility). Otherwise, the authors should refer to others who postulated a stable vs. flexible operating mode of executive functions. I am now aware of such a theoretical frame. 
  2. INTRODUCTION. Line 161-170 Here, two other crucial points must be clarified. While I agree that 'standard neuropsychological tests' could have several limits in assessing executive functions, I do not agree that psychophysical (behavioral) tests fail to assess executive functions' functioning/impairments. For instance, tasks such as the stop-signal or Go/No-go tasks are well suited to assess inhibitory control. Posner task can reliably assess selective attention as the Stroop task. Maybe a limit of most psychophysical tests is that they tend to assess the functioning of one executive function at a time (but see, Brown and Braver, 2005). Therefore, I suggest the authors say that their task provides a more holistic measure of how executive functions operate'.
  3. INTRODUCTION. Line 178 What ‘specific cognitive function are the authors referring to?
  4. METHODS Line 211 (and Line 559) What is the z score refereeing to? Usually, cognitive integrity is measured by the score of the MoCA or the MMSE. Is this a normalization of such values? Please clarify.
  5.  METHODS Line 267 In my view, the C3T (please define where the acronym comes from) assesses the switch from reactive to proactive mode.
  6. RESULTS Figure 3B. I guess the PA is reported on the Y-axis. Otherwise, you need to specify what score is represented. Delete PA from the X-axis.
  7. DISCUSSION Line 557 In fact, PD patients' performance does not worsen with age but with the disease stage. Please add this notion.
  8. DISCUSSION Line 563-570. At present, the best way to explain the differential (and not always positive effects) of dopaminergic therapy is the 'overdose dopamine hypothesis' (Cools, 2019). According to this hypothesis, drugs improve motor and cognitive functions relying on the depleted dorsolateral striatal circuit but overdose the relatively unaffected circuitry of the ventral striatum and associated prefrontal areas, impairing cognitive functions depending on these brain regions.
  9. DISCUSSION Lines 631-633 In fact, advanced Parkinson’s patients’ have an impairment in proactive control (e.g., Mirabella et al. 2017).

10.  DISCUSSION Line 638-650 Clarify since the beginning that this paragraph refers to healthy subjects.

MINOR POINTS

1.     Line 30. Provide a few examples

2.     Line 95 Delete inhibition, as just said, is a core executive function.

3.     Line 377 Remove the '_'

4.     Line 443 I guess, should be Table 3

5.     Line 528 Change to 'C3T'

6.     Lines 531-2 Change to 'Cognitive control is impaired in people with PD'. 

7.     Line 630 Do the authors mean PD patients with ICD? Please specify. 

8.     Lines 656-659 The next two sentences should be summarized as follows: 'This might be due to a larger variability and to a concomitant deterioration of cognitive in PD patients with respect to healthy people. In any case delete ‘(PD)’

 References

Braver TS (2012) The variable nature of cognitive control: a dual mechanisms framework. Trends Cogn Sci 16:106-113.

Brown JW, Braver TS (2005) Learned predictions of error likelihood in the anterior cingulate cortex. Science 307:1118-1121.

Cools R (2019) Chemistry of the Adaptive Mind: Lessons from Dopamine. Neuron 104:113-131.

Mirabella G (2023) Beyond Reactive Inhibition: Unpacking the Multifaceted Nature of Motor Inhibition. Brain Sciences 13.

Some passages should be further improved. Sometimes the language is convoluted and the sentences unnecessarily complex

Reviewer 3 Report

Authors revised the Manuscript according to the Reviewer's suggestion and it should be accepted in the current form

 A minor editing of English language is required for this Manuscript

Author Response

English language has been further edited. Thank you for your support and approval of the manuscript